# Transformative Trends in Runoff and Sediment Dynamics and Their Influential Drivers in the Wuding River Basin of the Yellow River: A Comprehensive Analysis from 1960 to 2020

Jingwei Yao [1,2], Zhanbin Li [1], Biao Zhu [3], Pan Zhang [2], Jingshu Wang [2], Weiying Sun [2], Shasha Mei [4], Yaqiang Zhang [5] and Peiqing Xiao [2,*]

1 State Key Laboratory of Eco-Hydraulics in Northwest Arid Region of China, Xi'an University of Technology, Xi'an 710048, China; solofromchina@163.com (J.Y.)
2 Key Laboratory of Soil and Water Conservation on the Loess Plateau of Ministry of Water Resources, Yellow River Institute of Hydraulic Research, Zhengzhou 450003, China
3 Hydrology Bureau of Yellow River Conservancy Commission, Zhengzhou 450004, China
4 School of Management Engineering, Zhengzhou University of Aeronautics, Zhengzhou 450046, China
5 Henan Water Conservancy Resettlement Affairs Center, Zhengzhou 450003, China
* Correspondence: 12022131378@stu.nxu.edu.cn; Tel.: +86-10-136-6385-8913

**Abstract:** The correlation between runoff and sediment challenges ecological preservation and sustainable development in the Yellow River Basin. An understanding of the key factors influencing variations in runoff and sediment transport in crucial river basins is essential for effective soil erosion management within the context of ecological and economic development. The Mann–Kendall test, Pettitt test, and Morlet wavelet analysis were employed in the Wuding River Basin to analyze the trends in runoff and sediment changes from 1960 to 2020. We explored the double cumulative curve method to assess the contribution rates of precipitation and human activities to the variability of runoff and sediment transport. We explored the primary factors driving the changes in runoff and sediment transport through random forest regression analysis. (1) From 1960 to 2020, annual precipitation in the Wuding River Basin increased minimally, while annual runoff and sediment transport decreased strongly with abrupt changes. Abrupt changes in annual runoff and sediment transport occurred in 1971 and 1979, respectively. (2) The relationship between runoff and sediment transport changed in approximately 1972 and 2000. The distribution of monthly runoff became more uniform during Periods II (1973–2001) and III (2002–2020) compared to that during the baseline period (1960–1972, Period I), while sediment transport became increasingly concentrated in the flood season. (3) During Period II, the contribution rates of climate and human activities to runoff and sediment transport were 11.94% and −14.5%, respectively, compared to the baseline period. During Period III, the contribution rates of climate and human activities to runoff and sediment transport were −11.9% and −17.7%, respectively. Human activities substantially reduced runoff and sediment, with greater impacts on sediment reduction. Climate weakly influenced basin sediment transport variations. (4) The normalized difference vegetation index (NDVI) and grassland area extent had the greatest impact on runoff, while the NDVI and forest area extent affected sediment transport.

**Keywords:** runoff–sediment correlation; sediment burden; anthropogenic interference; climatic perturbations; propelling mechanisms





## 1. Introduction

The dynamics of flow and sediment, an integral facet of surface material circulation, arise from the synergistic impacts of climatic factors and underlying surface characteristics, assuming a pivotal role in shaping and advancing regional water and soil resources [1,2]. Globally, climate change and anthropogenic endeavors have variably modified hydrological processes within watersheds, thereby influencing the stability of ecosystemic realms [3,4].

In response to the deteriorating ecological milieu, extensive ecological restoration initiatives have been undertaken worldwide since the 20th century, aiming to ameliorate the prevailing conditions [5–7]. Human-induced modifications on the Earth's surface have profoundly impacted the structure, functionality, and spatial arrangement of ecosystems, concurrently yielding alterations in the sediment and water processes within watersheds. In recent decades, significant changes have been observed in the flow and sedimentation of many rivers globally [8–10], driven primarily by climate variability and human activities. These changes exhibit significant spatiotemporal variations [11–13]. Consequently, enhancing our comprehension of the mechanisms and processes governing changes in watershed sediment and water conditions is of paramount significance to safeguard and comprehensively manage watershed ecosystems.

The Yellow River Basin holds significant ecological and economic importance in China, playing a crucial role in the strategic framework of social and economic development. Specifically, the middle reaches of the Yellow River exemplify an ecologically vulnerable area with severe soil erosion due to its high sediment content and coarse sand. Since 1950, the Loess Plateau has implemented large-scale measures to conserve soil and water, including afforestation, terracing, and the construction of check dams [14,15]. Notably, the Grain for Green Program, initiated in 1999, has brought about significant changes in the underlying landscape of the basin [2,16]. Human activities have exerted a profound influence on the water cycle and erosion sediment production in the Yellow River Basin. Monitoring data reveal a substantial decline of approximately 70% in the flow and sediment load of the Yellow River over the past 60 years. For instance, the sediment load at the Tongguan hydrological station on the main course of the Yellow River has dropped from an average of $16 \times 10^8$ tons per year in the 1970s to approximately $3 \times 10^8$ tons since 2000 [17]. Several studies have investigated the characteristics and driving mechanisms of water sediment evolution in the middle reaches of the Yellow River, with a consensus emerging that both flow and sediment load have significantly decreased, with human activities playing a more prominent role than climate change as the primary driver behind these declines [18].

The Wuding River, situated in the Loess Plateau, is a significant tributary in the middle reaches of the Yellow River, playing a substantial role in the hydrological and sedimentary dynamics of the Yellow River. Therefore, understanding the patterns and factors influencing water sediment changes in the Wuding River Basin is essential for comprehending runoff and sediment fluctuations in the Yellow River. The Wuding River Basin is a critical area for implementing soil and water conservation measures in the Loess Plateau and is also highly vulnerable to climate change. In recent years, a variety of methodologies, such as ecohydrological models, the elasticity coefficient method, regression analysis, and hydrological simulation, have been employed to analyze the relationship between runoff sediment changes and the response to climate and human activities in this region [19–21]. Previous studies have provided some insights into the erosion sediment production process and mechanisms in this basin. However, uncertainties still remain regarding the understanding of water sediment changes under the influence of human activities. Currently, there are conflicting viewpoints on the magnitude of changes in runoff and sediment load in the Wuding River Basin, as well as the extent of influence from various factors. Therefore, it is crucial to further clarify and quantitatively evaluate these aspects.

The study focused on the Wuding River Basin, a representative compound erosion area of wind and water in the middle reaches of the Yellow River. Multiple analysis methods were employed, including the M-K test, Pettitt test, Morlet wavelet analysis, double cumulative curve method, and random forest regression analysis. By analyzing long-term time series data of runoff and sediment as well as human activity data in the Wuding River Basin, the cycles and mechanisms of runoff and sediment variations in this basin were clarified. Additionally, a prediction index system for runoff and sediment was established. This system quantitatively identified the main natural and anthropogenic

factors influencing changes in runoff and sediment transport in the basin. Furthermore, the study explored the contribution rates of the causes and driving factors of runoff and sediment variations. The aim of this research was to provide a theoretical basis and data support for the comprehensive control of soil erosion in key ecologically vulnerable areas of the Yellow River Basin and the establishment of a scientific runoff and sediment regulation system at the basin level in China.

## 2. Study Area

The Wuding River is a primary tributary of the Yellow River, located in the southern edge of the Maowusu sandland and the northern part of the Loess Plateau. The main stream has a total length of 491.0 km, and the watershed covers an area of 30,261 km$^2$ (Figure 1). The Baijiachuan hydrological station, located in the lower reaches of the Wuding River, controls an area of 29,662 km$^2$, accounting for 98% of the total watershed area.

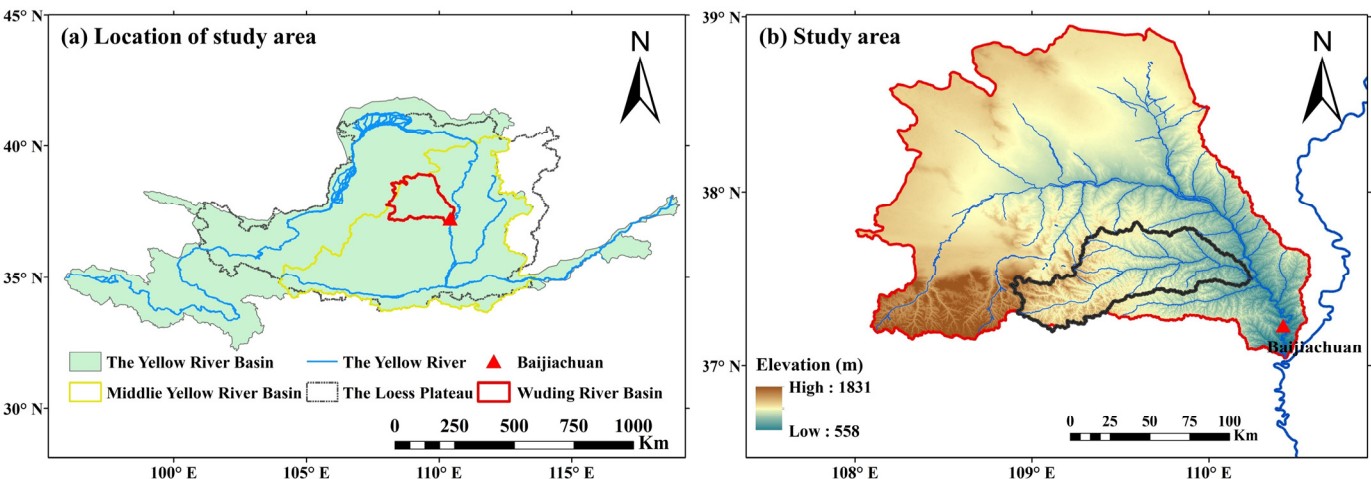

**Figure 1.** Location and characteristics of the studied watershed.

The Wuding River Basin belongs to the temperate continental arid and semiarid monsoon climate type, characterized by a dry climate, loose soil, and sparse vegetation. According to meteorological data from 1960 to 2020, the average annual precipitation in the Wuding River Basin is 475 mm. The distribution of precipitation within the year is extremely uneven, with over 80% of the rainfall occurring from May to September, mostly in the form of short-duration and high-intensity rainfall. This has resulted in severe soil erosion and intense sediment transport. The area of soil and water loss in the basin is 23,137 km$^2$, with an average erosion modulus of 6090 t/(km$^2$·a), making it one of the main sources of coarse sediment in the Yellow River.

The Wuding River Basin has a long history of comprehensive management of small watersheds. Since the initiation of soil erosion control work in 1950, the scope of basin management has been expanding. The 1950s to 1960s marked the initial stage, during which the management scope was limited, and a preliminary completion of 2153 km$^2$ was achieved. The 1970s to 1980s witnessed the stage of scaled-up management, focusing on channel improvement and the construction of numerous check dams. A total of 5929 check dams were built, accounting for 51.1% of the total number of check dams. In the 1980s, the Wuding River Basin was designated a key area for national soil and water conservation, and a series of comprehensive soil and water conservation measures were carried out with small watersheds as the management units. While intensifying management efforts, advanced practical techniques and achievements were actively promoted and applied. By 2010, the Wuding River Basin had constructed 2322.35 km$^2$ of farmland, 6234.68 km$^2$ of soil and water conservation forests, 975.18 km$^2$ of economic fruit trees, 1775.05 km$^2$ of grassland, and 360.70 km$^2$ of afforested areas. The forest and grass coverage rate reached 68.13%. Additionally, 7508 check dams, 19,628 reservoirs and wells, and 11,969 ponds and

bunds were constructed. According to the latest statistics, by 2018, the cumulative area of soil erosion control in the Wuding River Basin reached 12,996 km$^2$, achieving a soil erosion control rate of 50.93%.

## 3. Data and Methodology

### 3.1. Data Description

The annual runoff and sediment data from the Baijiachuan hydrological station at the outlet control station of the Wuding River Basin during 1960–2020 were obtained from the "Yellow River Basin Hydrological Yearbook". The daily meteorological data from four meteorological stations (Yulin, Hengshan, Suide, and Jingbian) within the basin during 1960–2020 were obtained from the China Meteorological Data Service Center (http://data.cma.cn (accessed on 15 March 2023)). The digital elevation model (DEM) of the basin with a spatial resolution of 30 m × 30 m was obtained from the Loess Plateau Scientific Data Center of the National Earth System Science Data Sharing Service Platform (http://loess.geodata.cn (accessed on 1 May 2023)). The normalized difference vegetation index (NDVI) used the GIMMS global vegetation index data provided by ECOCAST (https://poles.tpdc.ac.cn/en (accessed on 9 January 2023)) with a spatial resolution of 8 km and a temporal resolution of 15 days. The land use data for 1990–2020 were obtained from the Landsat-derived annual China land cover dataset (CLCD) [22]. The Wuding River Basin covers various counties and cities in Yulin city, Shaanxi Province. The population, GDP, and other socioeconomic statistics were sourced from the "Statistical Yearbook of Shaanxi Province" and the "Statistical Yearbook of Yulin City". The water consumption data for Yulin city were obtained from the "Water Resources Bulletin of Shaanxi Province" (2000–2015). The nighttime light remote sensing data product used was the "Prolonged Artificial Nighttime-light Dataset of China (PANDA)" published by the National Tibetan Plateau Data Center (https://data.tpdc.ac.cn/home (accessed on 2 February 2023)). This dataset covers the time series from 1984 to 2020 with a spatial resolution of 1 × 1 km and a temporal resolution of one year.

### 3.2. Methodology

This study employed mainstream research methods, such as the Mann–Kendall test, Pettitt test, Morlet wavelet analysis, and double mass curve method, for comparison with previous relevant research results. Additionally, the random forest regression model was used to analyze the primary driving factors influencing the variability of runoff and sediment transport in the Wuding River Basin. Through a comprehensive review of the literature related to runoff and sediment changes in the Wuding River Basin, this study represents the first application of this method in the region.

#### 3.2.1. Mann–Kendall Rank Correlation Trend Test

The Mann–Kendall rank correlation trend test method (hereinafter referred to as the M-K test) is a widely used and effective nonparametric statistical method for analyzing trends in time series data. It is commonly employed in the trend analysis of hydrological, meteorological, and other time series due to its simplicity, robustness, and high level of quantification [23–27]. This method disregards the distribution characteristics of the sample series and is unaffected by a few extreme values, making it suitable for detecting trends in time series [28].

For a time series variable $(X_1, X_2, \ldots, X_n)$, where $n$ represents the length of the time series, the M-K test introduces a statistical parameter denoted as $S$.

$$S = \sum_{k=1}^{n-1} \sum_{j=k+1}^{n} \text{sgn}(x_j - x_k) \tag{1}$$

where sgn() is the sign function and is shown as follows:

$$\text{sgn}(x_j - x_k) = \begin{cases} 1 & x_j - x_k > 0 \\ 0 & x_j - x_k = 0 \\ -1 & x_j - x_k < 0 \end{cases} \tag{2}$$

When $S$ is a normal distribution, then the variance $Var(S) = n(n-1)(2n+5)/18$. For values of $n > 10$, the statistical measure of the normal distribution is:

$$Z_{MK} = \begin{cases} \frac{S-1}{Var(S)} & S > 0 \\ 0 & S = 0 \\ \frac{S+1}{Var(S)} & S < 0 \end{cases} \tag{3}$$

The trend is examined using the $Z_{MK}$ value. A positive (negative) value of $Z_{MK}$ indicates an upwards (downwards) trend in the series being tested. When the absolute value of the test statistic is greater than 1.28, 1.64, and 2.32, the series passes the significance test at the 90%, 95%, and 99% confidence levels, respectively. The larger the absolute value of the test statistic is, the more significant the trend of the series is.

### 3.2.2. Theil–Sen Median Estimator Linear Model

The Theil–Sen median estimator linear model, also referred to as Sen's slope estimator, is a robust nonparametric statistical technique utilized for calculating trends. This method is computationally efficient and insensitive to measurement errors and outliers, making it well suited for the trend analysis of lengthy time series data [29,30]. The Theil–Sen model is utilized to determine the slope of the series, denoted as $\beta$. The slope $\beta$ represents the average rate of change and trend in the time series. A positive value of $\beta$ indicates an upward trend, while a $\beta$ value of 0 suggests an insignificant trend. Conversely, a negative $\beta$ value indicates a downward trend. The calculation formula for Sen's slope of a time series $x_t = (x_1, x_2, \ldots, x_n)$ is as follows:

$$\beta = M_f \left( \frac{x_j - x_i}{j - i} \right), \forall j > i \tag{4}$$

where $M_f$ is the median function.

### 3.2.3. Pettitt Change-Point Test

The Pettitt change-point test is used to determine whether there is a significant change point in a hydrometeorological time series, even when the exact timing of the change is unknown. For a hydrometeorological time series $X = (x_1, \ldots, x_n)$, assuming the change point occurs at $X_t$, the original time series can be divided into two parts: $x_1, x_2, \ldots, x_t$ and $x_{t+1}, x_{t+2}, \ldots, x_n$. The statistic $U_{t,n}$ is defined to assess the possible occurrence of a change point at time $t$:

$$U_{t,n} = U_{t-1,n} + \sum_{j=1}^{n} \text{sgn}(x_t - x_j) \ t = 2, \ldots, n \tag{5}$$

$U_{i,n} = \sum\limits_{j=1}^{n} \text{sgn}(x_i - x_j)$ and $\text{sgn}(\cdot)$ denote the sign function, which is calculated according to Equation (2).

To determine the probable occurrence time, $t$, of a mutation point, the statistical measure $K_t$ is defined to locate the most likely mutation point.

$$K_t = \max_{1 \le t \le n} |U_{t,n}| \tag{6}$$

After identifying the mutation point using Equation (6), the significance level $P_t$ is calculated using the following formula:

$$P_t = 2 \exp\left(\frac{-6K_t^2}{n^3 + n^2}\right) \tag{7}$$

For a given confidence level $\alpha$, if $P_t > \alpha$, the null hypothesis is accepted, indicating no significant mutation at time $t$; if $P_t < \alpha$, the null hypothesis is rejected, indicating a significant mutation at time $t$. In this paper, a confidence level of $\alpha = 0.5$ was chosen.

3.2.4. Double Mass Curve

The double mass curve method (DMC) is commonly used to examine the consistency and evolution trends of hydrological elements [31]. It plots the cumulative values of one variable over a specific period against the corresponding cumulative values of another variable on a Cartesian coordinate system. The cumulative relationship curve is analyzed to determine the changing trends, timing, and magnitude of the response relationship between the two variables. The double cumulative curve of runoff and sediment transport can be used to study the abrupt changes and trend intensity of sediment transport caused by human activities in rivers [32]. If the slope of the cumulative curve significantly deviates, it indicates that human activities have a significant impact on the changes in runoff and sediment transport. The year corresponding to the deviation point is identified as the year of the abrupt change, and the greater the deviation in slope is, the higher the degree of human interference is [33].

The curve can be represented as two variables, $x(t)$ and $y(t)$. Within a certain observation period length $t$, there are observed values $x_i(t)$ and $y_i(t)$, where $i = 1, 2, 3, \ldots, n$. The cumulative values of the time series for the variables $x(t)$ and $y(t)$ are calculated as $x'_i(t)$ and $y'_i(t)$, respectively, using the following formulas:

$$x'i(t) = \sum_{i=1}^{n} xi(t), \quad i = 1, 2, \ldots, n \tag{8}$$

$$y'i(t) = \sum_{i=1}^{n} yi(t), \quad i = 1, 2, \ldots, n \tag{9}$$

The cumulative values of the two variables are plotted on the $x$-axis and $y$-axis of a Cartesian coordinate system, resulting in a double cumulative curve. In this study, the cumulative annual runoff is plotted on the $x$-axis, and the cumulative annual sediment transport is plotted on the $y$-axis, resulting in a graph of the cumulative curve of runoff and sediment transport.

3.2.5. Copula Function

The theoretical foundation of the copula function lies in the decomposition of an $n$-dimensional joint distribution function into $n$ marginal distribution functions and a copula function that describes the dependency structure between variables. It can be expressed as follows: let $X = (x_1, x_2, \ldots, x_n)$ be an $n$-dimensional random variable with marginal distribution functions $F_1, F_2, \ldots, F_n$; then, the $n$-dimensional copula function $C$ satisfies the following formula:

$$H(x_1, x_2, \ldots, x_n) = C[F_1(x_1), F_2(x_2), \ldots, F_2(x_n)] \tag{10}$$

Function $C$ in the equation represents a joint distribution function that is uniformly distributed in the interval [0, 1]. In hydrological frequency analysis and calculations, commonly used copula functions are the elliptical and Archimedean types (Table 1). Elliptical copula functions have elliptical contour line distributions and are not limited by the dependence structure between variables. Archimedean copula functions, on the other hand,

have a simpler form and strong symmetry. In this study, we selected three single-parameter distributions from the most commonly used Archimedean copula functions in hydrological research [34–36], namely, the Gumbel copula, Clayton copula, and Frank copula, to fit the joint distribution function. The fitting performance of these three copula functions was evaluated using the squared Euclidean distance ($d_2$) and the Akaike information criterion (AIC method) for model selection.

**Table 1.** Expression of copulas and their parameters.

| Classification | Names | Distribution Functions and Parameters |
|---|---|---|
| Elliptical | Gaussian | $C(u,v) = \Phi_\theta\left(\Phi^{-1}(u), \Phi^{-1}(v)\right)$, $\theta$ represents a parameter |
| | Student's t | $C(u,v) = t_{\theta,k}\left(t_k^{-1}(u), t_k^{-1}(v)\right)$, $\theta$ represents a parameter, $k$ represents the degrees of freedom |
| Archimedean | Gumbel | $C(u,v) = \exp\left\{-\left[(-\ln u)^\theta + (-\ln v)^\theta\right]^{-\frac{1}{\theta}}\right\}$, $\theta \geq 1$ |
| | Clayton | $C(u,v) = \left(u^{-\theta} + v^{-\theta} - 1\right)^{-\frac{1}{\theta}}$, $\theta > 0$ |
| | Frank | $C(u,v) = -\frac{1}{\theta}\ln\left[1 + \frac{\left(e^{-\theta u}-1\right)\left(e^{-\theta v}-1\right)}{e^{-\theta}-1}\right]$, $\theta \neq 0$ |

### 3.2.6. Continuous Wavelet Analysis

Wavelet analysis, a powerful statistical tool, was originally utilized in the field of signal processing and analysis and is currently widely applied in various disciplines, including hydrology and ecosystems [37–39]. Wavelet transformation can be categorized into continuous wavelet transform (CWT) and discrete wavelet transform (DWT). Among these, the continuous wavelet transform is particularly suitable for feature extraction. The equation for the continuous wavelet transform is shown as follows:

The continuous wavelet transforms of a discrete time series $x_n (n = 1, \ldots, N)$ with equal time steps $\delta_t$ are defined as the convolution of the wavelet function $x_n$ under scale and translation.

$$W_n^X(s) = \sqrt{\frac{\delta_t}{s}} \sum_{n'=0}^{N-1} x_n' \psi * \left[\frac{(n'-n)\delta_t}{s}\right] \tag{11}$$

In the equation, the symbol * represents the complex conjugate, and $N$ represents the total number of data points in the time series. $(\delta_t/s)^{1/2}$ is a factor used to normalize the wavelet function, ensuring that the wavelet function has unit energy at each wavelet scale $s$. By transforming the wavelet scale $s$ and localizing it along the time index $n$, a waveform can be obtained that displays the fluctuation characteristics of the time series at a certain scale and their variation over time, which is known as the wavelet power spectrum.

When conducting a wavelet transformation on a time series, the choice of the mother wavelet is highly important. Typically, when analyzing a time series, it is desirable to obtain smooth and continuous wavelet amplitudes, making nonorthogonal wavelet functions more suitable. Additionally, to obtain information regarding both the amplitude and phase aspects of the time series, complex-valued wavelets should be selected, as complex-valued wavelets have an imaginary component that can effectively express the phase [40]. The Morlet wavelet not only possesses nonorthogonality but also is an exponential complex-valued wavelet modulated by a Gaussian function, with its equation expressed as follows:

$$\psi_0(t) = \pi^{-1/4} e^{i\omega_0 t} e^{-t^2/2} \tag{12}$$

In the equation, $t$ represents time and $\omega_0$ is a dimensionless frequency. When $\omega_0 = 6$, the wavelet scale $s$ is approximately equal to the Fourier period ($\lambda$, $\lambda$ = 1.03 s), allowing for interchangeability between the scale and period terms. This balance in localization between time and frequency is a key feature of the Morlet wavelet. It is evident that the Morlet wavelet contains additional oscillatory information, as the wavelet power can encapsulate

both positive and negative peaks within a broad peak. In this study, the Wavelet toolbox in MATLAB software (Version: R2022a, MathWorks, Portola Valley, CA, USA) was utilized to calculate the wavelet coefficients for streamflow and sediment transport. The resulting wavelet transform real part distribution and variance plots were generated using OriginPro software (Version: 2021b, OriginLab, Northampton, MA, USA).

### 3.2.7. Random Forest Model

The random forest model is a widely used nonparametric regression machine learning algorithm in various fields. It consists of decision trees, which contribute to its high accuracy. To address overfitting, the random forest model utilizes a random sampling method with replacement when selecting feature attributes from the dataset [41]. The determination of the number of decision trees and randomly selected features is based on the model's fitting performance during the benchmark period. Ultimately, the best model is selected based on the coefficient of determination and the Nash efficiency coefficient. The basic form of the random forest model can be expressed as follows:

$$g_i = \frac{1}{n}\sum_{i=1}^{n} h_i(m_i) \tag{13}$$

In the equation, $g_i$ represents the predicted value, $n$ represents the total number of samples, $h_i(m_i)$ represents the training function for each decision tree, and $m_i$ represents the measured data.

This study selects the Bootstrap resampling method for training set sample processing. According to the rules, the sample set is divided into two parts and the decision tree model is built using the binary recursive method. Each decision tree in the model is independent and does not interfere with other trees. Finally, the well-grown decision trees are combined to obtain a classifier, namely, random forest. The classification results of the model data are determined by voting to determine the category of new samples, in order to achieve data prediction. A decision tree is a typical single classifier, which deduces the classification rules of decision trees for the classification of data from a large and disorderly set of samples by the recursive method, and then it analyzes data using these rules. This study uses the Classification and Regression Tree (CART) to implement the node splitting of the decision tree. The CART algorithm measures the data partition standard based on the *Gini* index and uses the feature value with the smallest *Gini* index as the splitting attribute of the node, which can explain the generated rules.

The calculation formula for the *Gini* coefficient is as follows:

$$Gini(T) = 1 - \sum_{i=1}^{k} [p(i|t)]^2 \tag{14}$$

In the equation, $T$ represents the sample; $k$ represents the number of categories in the sample; and $p(i|t)$ represents the probability of category $i$ at node $t$.

Calculate the index of the partition:

$$Gini(T) = \sum_{i=1}^{m} \frac{n_i}{n} Gini(i) \tag{15}$$

In the equation, $m$ represents the number of child nodes; $n_i$ represents the number of samples at child node $i$; and $n$ represents the number of samples at the parent node.

During the attribute splitting process, the CART algorithm calculates the parameters, namely, the *Gini* coefficient, based on Formulas (14) and (15). According to the calculation results, the priority attribute for node splitting is selected, which is the attribute with the smallest *Gini* coefficient. Through recursive iteration, the decision tree is continuously updated until a complete decision tree is generated.

Random forest regression analysis enables the ranking of the relative importance of influencing factors, thereby identifying the main controlling factors that significantly affect variations in runoff and sediment transport. This process facilitates the prediction of streamflow and sediment transport. The importance of variables is determined by the increase in the prediction error resulting from permuting the variable data. The percentage reduction in prediction error (IncMSE) serves as an indicator, measuring the extent to which the accuracy of random forest predictions diminishes when a variable's value is changed to a random number. A higher IncMSE value indicates a greater importance of the corresponding influencing factor and a higher contribution to the model [42,43].

## 4. Results

### 4.1. Runoff and Sediment Variation Characteristics

The trends of precipitation, annual runoff, and annual sediment transport at the Baijiachuan hydrological station during the period from 1960 to 2020 were examined through the application of the M-K test (Figure 2, Table 2). The outcomes of the statistical analyses revealed that there was relatively limited interannual variability in precipitation. However, both annual runoff and sediment transport exhibited substantial fluctuations on an interannual basis. Over the span of the past six decades, the average multiyear precipitation in the Wuding River Basin amounted to 475 mm, with the highest recorded value reaching 698 mm and the lowest reaching 264 mm. The annual precipitation demonstrated a slight upwards trend, although it did not pass the threshold for statistical significance. Sen's slope analysis indicated that the annual precipitation experienced a gradual increase at a rate of 1.25 mm·a$^{-1}$. The multiyear average runoff at the Baijiachuan hydrological station stood at $10.55 \times 10^8$ m$^3$, with the maximum value recorded at $20.15 \times 10^8$ m$^3$ and the minimum at $6.086 \times 10^8$ m$^3$. The annual runoff, however, exhibited a significant decreasing trend, with an average reduction of $0.12 \times 10^8$ m$^3$·a$^{-1}$. The multiyear average sediment transport at the Baijiachuan hydrological station was $0.81 \times 10^8$ t, with the maximum recorded value at $3.75 \times 10^8$ t and the minimum at $0.0267 \times 10^8$ t. The annual sediment transport showed an average decline of $0.02 \times 10^8$ t·a$^{-1}$, and this downwards trend was highly significant at the 0.001 level of significance. Since the 1970s, when soil and water conservation measures in the watershed began to take effect, sediment transport in the basin has gradually diminished as a result of the impact of these measures and reduced water supply.

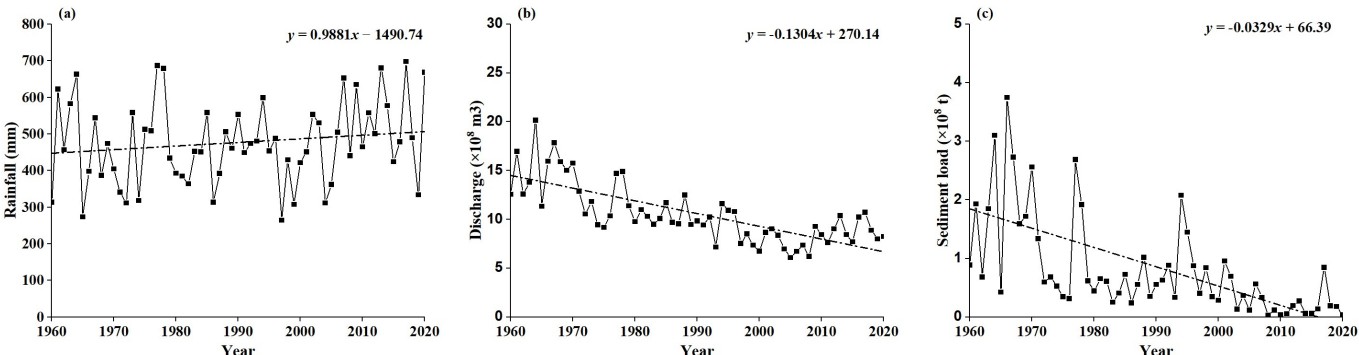

**Figure 2.** Temporal variations in annual precipitation (**a**), streamflow (**b**), and sediment transport (**c**).

**Table 2.** M-K test of annual precipitation, streamflow, and sediment transport for 1960–2010.

| Characteristic Value of Runoff and Sediment | $Z_{MK}$ | Sen's Slope |
|---|---|---|
| Annual rainfall | 1.21 | 1.25 |
| Annual discharge | −6.67 *** | −0.12 |
| Annual sediment load | −6.14 *** | −0.02 |

Note: *** denotes that the value of Z is significant at the 0.001 level.

Pettitt tests revealed noteworthy change points ($p < 0.001$) for both the yearly runoff and sediment load at the Baijiachuan hydrological station (Table 3). The transition point for annual runoff transpired in 1971, while the transition point for sediment load transpired in 1979. These transition points align with the era of rigorous soil and water conservation measures implemented on the Loess Plateau, thus validating the outcomes of the transition point analysis. By contrasting the interannual and intra-annual fluctuations in runoff and sediment load before and after the transition points in the Wuding River Basin, the following observations were deduced: (I). The average runoff for 1960–1979 and 1980–2020 in the Wuding River Basin was 523.62 m$^3$ and 488.83 m$^3$, respectively, denoting a 33.9% decline in runoff from 1980 to 2020 in comparison to 1960–1979. (II). The average sediment loads for 1960–1971 and 1972–2020 in the Wuding River Basin were $1.88 \times 10^8$ t and $0.55 \times 10^8$ t, respectively, signifying a notable 70.7% decrease in sediment load from 1972 to 2020 in comparison to that of 1960–1971. The reduction in sediment load surpassed the reduction in runoff.

**Table 3.** Pettitt test of the change point of annual discharge and sediment load.

| Statistical Parameter | Annual Discharge | Annual Sediment Load |
|---|---|---|
| Change-point year | 1979 *** | 1971 *** |
| Prechange year | $13.66 \times 10^8$ m$^3$ | $1.88 \times 10^8$ t |
| Postchange year | $9.03 \times 10^8$ m$^3$ | $0.55 \times 10^8$ t |
| Relative change | −33.9% | −70.7% |

Note: *** denotes that the Pettitt test results are significant at the 0.001 level.

By utilizing Morlet continuous wavelet analysis, the temporal variation in runoff and sediment data in the basin was assessed to identify the underlying cycles of runoff and sediment changes. The wavelet coefficients and wavelet variances in annual runoff and sediment transport at the Baijiachuan hydrological station are depicted in Figure 3. The diagram shows two extensive alternations of wet and dry periods in the annual runoff and sediment transport at the Baijiachuan hydrological station. The annual runoff predominantly displays three "wet-dry" cycles: 3 years, 9 years, and 28 years. Among them, the 28-year cycle of annual runoff had relatively stable behavior throughout the entire time domain, and the peak of wavelet variance transpired at this time scale, indicating that the primary period of annual runoff variation at the Baijiachuan hydrological station was 28 years. The annual sediment transport primarily exhibits three cycles: 2 years, 13 years, and 30 years. Among them, the 30-year cycle of annual sediment transport had relatively stable behavior throughout the entire time domain, and the peak of sediment transport variance transpired at the 30-year scale, indicating that 30 years was the dominant period of annual sediment transport variation at the Baijiachuan hydrological station. The temporal characteristics of annual runoff and sediment transport sequences bore similarities; however, there were also disparities, with the sediment transport cycle slightly surpassing the runoff cycle.

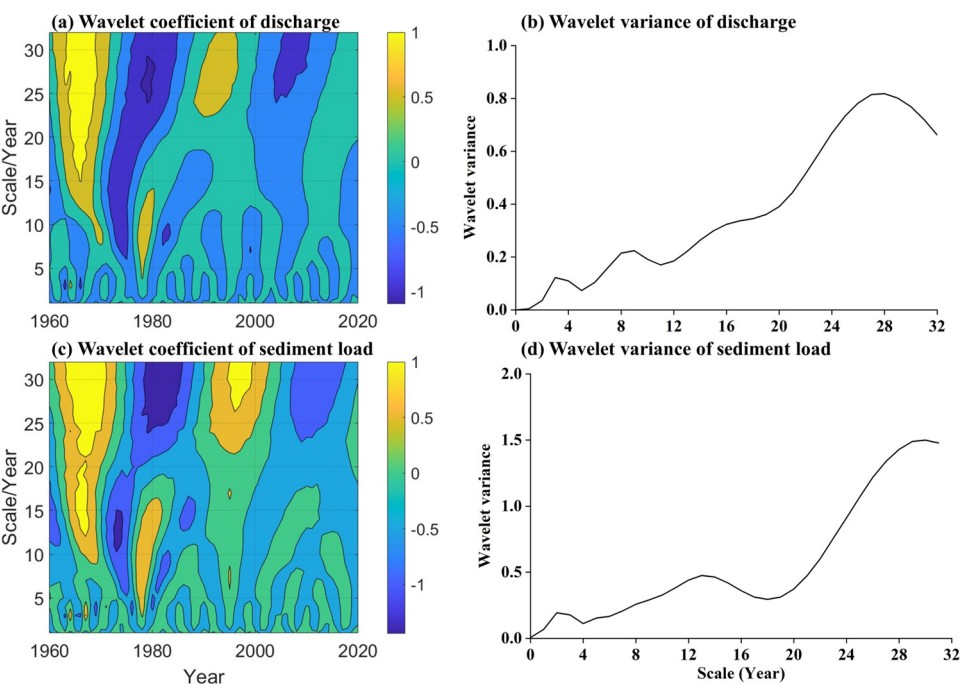

**Figure 3.** Coefficients and variance in annual runoff and sediment transport at the Baijiachuan Hydrological Station from 1960 to 2020.

### 4.2. Diagnosing the Variability in the Sediment–Discharge Relationship

Based on the aforementioned analysis findings, the reference period was established from 1960 to 1972 (Period I), followed by 1973 to 2001 (Period II), and culminating with 2002 to 2020 (Period III). The cumulative sediment–discharge curve at the Baijiachuan hydrological station exhibits a distinct convex form (Figure 4), signifying that the fluctuations in runoff and sediment within this basin occurred asynchronously. The extent of sediment attenuation surpassed that of runoff, with a discernible decline in sediment concentration. Furthermore, the cumulative curve manifests inflection points for approximately 1972 and 2000. The findings of relevant studies [20] also indicate a significant decrease in annual runoff in the Wuding River Basin from 1996 to 2007, with a sudden change occurring in 1971. Prior to the abrupt change, the trend in runoff was not significant, but after the change, a significant decrease in runoff was observed. These results are generally consistent with the findings of this study. In the 1970s, large-scale sediment control interventions, including silt dams and terraces, were implemented in the Wuding River basin, resulting in notable reductions in sediment. The cumulative sediment–discharge curve during Period II deviates significantly from Period I, showcasing a more substantial decrease in sediment load in comparison to runoff. After 2000, despite an augmentation in precipitation and subsequent runoff, the implementation of land conversion initiatives to foster forests and grasslands further curtailed sediment yield, leading to a subsequent downwards shift in the cumulative curve.

The period spanning from 1960 to 1972 was utilized as the reference point, and a comparative analysis was conducted on the variations in monthly average runoff and sediment discharge between the intervals of 1973 to 2001 and 2002 to 2020 (Figure 5). In comparison to Period I, both Period II and Period III displayed varying degrees of reduction in the multiyear average monthly runoff, particularly in March and July to September. Furthermore, a discernible trend towards a more equitable distribution of monthly runoff throughout the year was observed, with the proportion of flood season runoff declining from 41.5% in the baseline period to 37.9% in Period II and further declining to 35.9% in Period III. In stark contrast, when the mean monthly sediment discharge between the periods of 1960–1971 and 1972–2020 were contrasted, a substantial decline in sediment discharge was evident for each month; however, the pattern diverged from that of runoff.

Sediment discharge exhibited a greater concentration during the flood season, with the proportion of flood season sediment discharge escalating from 90.4% in Period I to 97.6% in Period II.

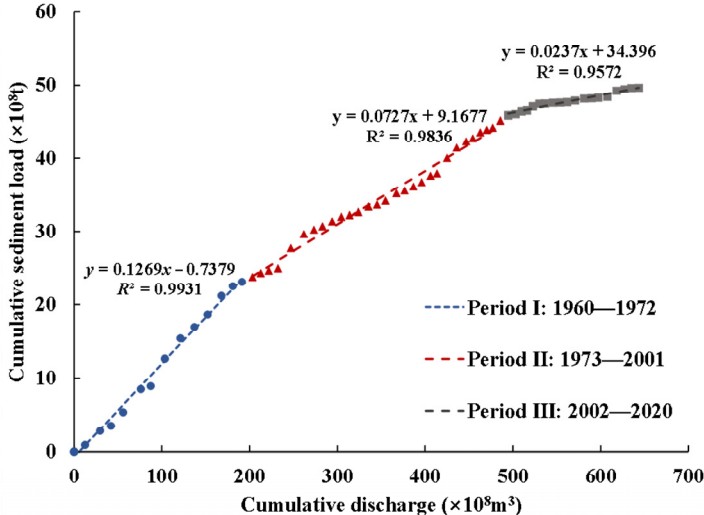

**Figure 4.** Cumulative curves of annual runoff and sediment discharge at Baijiachuan hydrological station.

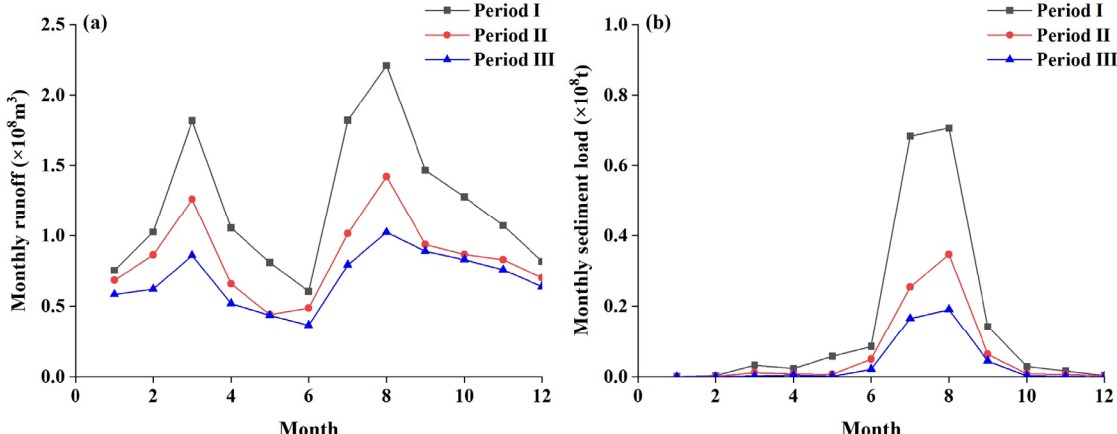

**Figure 5.** Analysis of the annual distribution characteristics of runoff (**a**) and sediment discharge (**b**) at Baijiachuan hydrological station.

The co-occurrence of runoff and sediment discharge was assessed through the utilization of the widely employed Kolmogorov–Smirnov (K-S) test and Akaike information criterion (AIC) (Table 4). The K-S test yielded *p* values exceeding 0.05, affirming that the fitting outcomes successfully met the test criteria. Consequently, only the AIC test outcomes are presented in the table to further determine the optimal joint distribution. Table 4 shows that the joint distribution of runoff–sediment discharge at the Baijiachuan hydrological station during Periods I, II, and III adheres to Gaussian, Gumbel, and Clayton distributions, respectively. This finding underscores the dynamic nature of the runoff–sediment relationship, as the joint distribution undergoes alterations. The correlation between runoff and sediment discharge during Period I exhibited a significantly greater magnitude than that of Period II and Period III. Transitioning from Period I to Period III, the correlation coefficient between annual runoff and sediment discharge demonstrated a continuous decrease, which is particularly pronounced in Period III. This observation suggests that the hydrosediment relationship within the watershed becomes increasingly intricate amidst intensified human activities.

**Table 4.** Goodness-of-fit test of five candidate copulas according to AIC.

| Stage | Spearman Coefficient | Gaussian | Student-T | Gumbel | Clayton | Frank |
|-------|---------------------|----------|-----------|--------|---------|-------|
| Period I | 0.90 | −17.53 | −15.36 | −13.94 | −16.21 | −17.43 |
| Period II | 0.72 | −12.91 | −11.01 | −16.94 | −4.91 | −13.12 |
| Period III | 0.41 | −5.81 | −3.66 | −2.82 | −1.23 | −8.37 |

*4.3. The Response of Runoff and Sediment to Climate and Human Activities*

Table 5 demonstrates a gradual intensification of the impact of human activities on runoff and sediment discharge in the Wuding River Basin from Period I to Period III. This trend coincides with the implementation of extensive soil and water conservation measures in the basin, including the construction of silt dams during the 1970s and the initiation of the Grain for Green program in 1999. These measures have exerted a profound impact on the hydrological and sedimentary dynamics within the basin. As these management practices expanded in scale, they brought about changes in surface morphology, consequently exerting a significant influence on the processes of runoff generation and sediment production. In comparison to Period I, the cumulative reductions in runoff and sediment discharge during Period II in the Wuding River Basin amounted to $4.6 \times 10^8$ m$^3$ and $1.02 \times 10^8$ t, respectively. Climate factors accounted for 11.94% and −14.5% of the variations observed in runoff and sediment discharge, respectively, while human activities contributed 88.06% and 114.5%, respectively. Remarkably, during this period, there was a substantial decrease in runoff and sediment discharge despite there being no significant decline in precipitation. Moving into Period III, the total declines in runoff and sediment discharge in the Wuding River Basin reached $6.4 \times 10^8$ m$^3$ and $1.55 \times 10^8$ t, respectively. Climate factors contributed −11.9% and −17.7%, while human activities accounted for 111.9% and 117.7% of the variations. Human activities emerged as the primary driver behind the reduction in sediment and water, with a more pronounced impact on sediment discharge, whereas the influence of climate change on sediment discharge within the basin remained relatively weak. The results of relevant studies [21,33] indicate that after 2001, due to the implementation of policies such as land reclamation and afforestation, human activities have intensified. Although there has been an increase in rainfall in the Wuding River Basin since 2007, the runoff and sediment transport continue to decrease. From 2001 to 2020, human activities contributed to over 110% of the reduction in runoff and sediment. Measures like check dams, land reclamation and afforestation, and reservoir construction and irrigation are among the factors causing the decrease in runoff and sediment in the Wuding River Basin, with check dams and reservoirs playing a major role in reducing sediment transport. These findings are consistent with the results of this study.

**Table 5.** The contributions of climate and human activities to changes in runoff and sediment discharge.

| Type | Time | Observation | Decrement | Contribution Rate from Climate Change/% | Contribution Rate from Human Activity/% |
|------|------|-------------|-----------|------------------------------------------|------------------------------------------|
| Runoff | Period I | $14.7 \times 10^8$ m$^3$ | - | - | - |
| | Period II | $10.1 \times 10^8$ m$^3$ | $4.6 \times 10^8$ m$^3$ | 11.94 | 88.06 |
| | Period III | $8.3 \times 10^8$ m$^3$ | $6.4 \times 10^8$ m$^3$ | −11.9 | 111.9 |
| Sediment Load | Period I | $1.78 \times 10^8$ t | - | - | - |
| | Period II | $0.76 \times 10^8$ t | $1.02 \times 10^8$ t | −14.5 | 114.5 |
| | Period III | $0.23 \times 10^8$ t | $1.55 \times 10^8$ t | −17.7 | 117.7 |

The impact of human activities on the alterations in runoff and sediment within river basins is intricate and diverse. The influence of human activities on runoff primarily stems from the modification of land use, which in turn affects climate and underlying surface conditions. Conversely, the effect of human activities on sediment transport mainly arises

from measures such as the construction of check dams and the restoration of vegetation. Therefore, it is imperative to deeply study the human activities that instigate changes in runoff and sediment to effectively manage soil erosion and achieve the objectives of water conservation, soil preservation, and the mitigation of runoff and sediment transport.

In this study, we meticulously selected 10 indicators that holistically depict the intensity of human activities, encompassing the arable land area, GDP, population, NDVI, forest area, grassland area, water area, nighttime light remote sensing, watershed soil erosion control area, and irrigation water consumption, through an extensive literature review and yearbook consultations. We subsequently investigated the driving impact of human activities on changes in runoff and sediment within the Wuding River Basin. Due to the insufficiency of pre-1990 data pertaining to human activity factors, we constructed a random forest model utilizing runoff and sediment transport as dependent variables for the period of 1990–2015, with human activity indicators serving as independent variables to ensure temporal consistency.

To evaluate the accuracy of the fitted values obtained from the random forest regression model, we computed the $R^2$ value and mean absolute error (MAE). The analysis and validation of the predicted values derived from the random forest regression model, in comparison with the measured values, are presented in Figure 6. Notably, both the $R^2$ values of the fitted curves for annual runoff and sediment transport from the random forest regression model and the measured values reached 0.8, indicating a close approximation between the fitting results and the actual values, thus asserting their reliability.

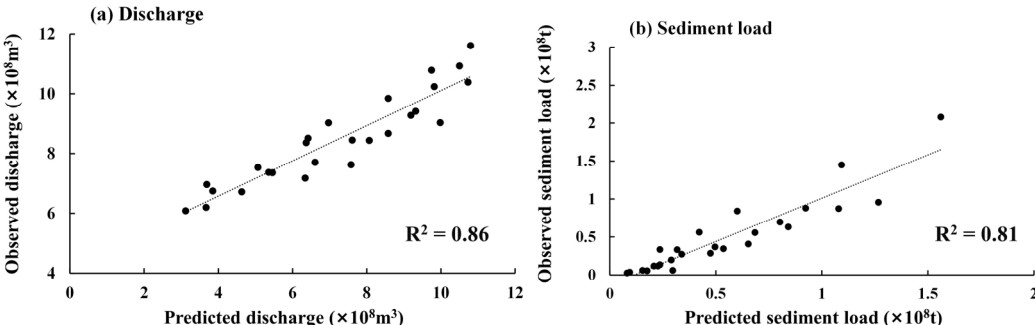

**Figure 6.** The fitting and validation of the random forest model.

The runoff and sediment transport data were used to fit a random forest regression model. To discern the relative importance of each human activity factor in influencing runoff and sediment transport, the percentage reduction in prediction error (IncMSE) was plotted for the 10 factors, as illustrated in Figure 7. The findings revealed that the NDVI and grassland area had the most profound impact on runoff, with IncMSE values of 14.63% and 14.26%, respectively. Following closely were the forest area (6.90%) and arable land area (5.93%). In terms of sediment transport, the NDVI and forest area emerged as the most significant human activity factors, with IncMSE values of 15.93% and 14.97%, respectively. These were trailed by the soil erosion control area (6.48%) and grassland area (6.07%).

Notably, the NDVI exhibited a significant negative correlation with both runoff and sediment transport, while the arable land area displayed a positive correlation with runoff. The continuous proliferation of forest and grass vegetation intercepted more precipitation, subsequently impeding the flow rate through the interception of dense litter. Moreover, a small portion of the precipitation replenished the groundwater, leading to a consistent decrease in runoff. This perpetual increase in water storage and soil conservation effects attributed to vegetation culminated in a more pronounced reduction in sediment transport.

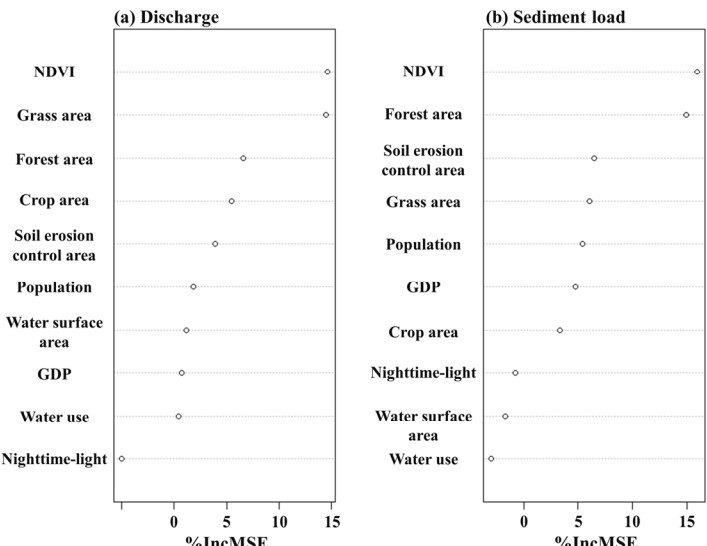

**Figure 7.** Ranking the importance of factors influencing annual runoff and sediment transport.

## 5. Discussion

### 5.1. Variation in Runoff and Sediment in the Wuding River Basin and a Comparative Analysis of Their Influencing Factors

This study unveils noteworthy inflection points in the annual runoff and sediment discharge in the Wuding River Basin over the past six decades, with 1971 marking the change in runoff and 1979 marking the change in sediment discharge. Human activities play a dominant role in diminishing both runoff and sediment, with a greater impact observed on sediment discharge reduction. These findings are consistent with the majority of previous studies [19,21,33]. Previous research has commonly employed attribution analysis and other methodologies to examine natural factors, including climate and precipitation, as well as human factors, such as afforestation and check dam construction, which influence soil erosion. The results demonstrate that human factors play a leading role in the variations in runoff and sediment in the Wuding River Basin. Among these factors, irrigation and afforestation primarily contribute to runoff reduction, while afforestation, grass planting, and check dam construction aid in sediment reduction. The NDVI and cumulative annual check dam control area are significantly negatively correlated with sediment transport. Moreover, the cumulative annual check dam control area has a greater direct impact on sediment transport than the NDVI. It is identified as the primary factor influencing sediment transport variability. Further research indicates that the annual variability of sediment transport in the Wuding River Basin is jointly driven by check dam construction and NDVI changes. The mean jump variability is mainly influenced by precipitation and check dam construction changes. Variance variability is primarily caused by changes in precipitation and NDVI. For annual runoff, trend variability is driven by changes in evaporation, check dam construction, reservoir construction, and agricultural irrigation water consumption. Mean jump variability is mainly influenced by changes in precipitation, reservoir construction, and agricultural irrigation water consumption. Variance variability is primarily caused by changes in precipitation.

This study also identifies the factors with the most significant impact on runoff, such as the NDVI and grassland area, and the most substantial human activity factors that affect sediment discharge, such as the NDVI and forest area. Related studies suggest [44,45] a close negative correlation between annual runoff in the Wuding River Basin and the extent of soil and water conservation measures. The contributions of maximum monthly cumulative precipitation, maximum daily precipitation, annual precipitation, and weighted average of the area of soil and water conservation measures to the variation in annual runoff in the Wuding River Basin are 37.5%, 26.9%, 9.4%, 14.5%, and 11.8%, respectively.

Based on relevant reports on sediment accumulation in the Wuding River Basin, it has been found that there is localized sediment accumulation in some sections due to floods, but there is no large-scale, long-lasting sediment accumulation. This finding suggests a direct relationship between the decrease in sediment production and the reduction in sediment discharge in the Wuding River Basin [46]. In comparison with other research findings, this study reveals that the erosion area and high-intensity erosion area in the Wuding River Basin are still larger than those in the Jing River, Beiluo River, Wei River, and Fen River in the middle reaches of the Yellow River. However, the rate of reduction in the soil erosion area is significantly higher than that in the middle reaches of the Yellow River, especially in the Huangfuchuan River, Jialu River, Jiaohe River, Qiushui River, and Beiluo River, which are key tributaries in the area with high sand and coarse sediment contents. Therefore, the current state of soil erosion and the effectiveness of soil and water conservation in the Wuding River Basin have significant implications for conducting soil and water conservation work in the middle reaches of the Yellow River.

*5.2. Limitations of This Study*

Due to certain constraints, specifically the limited duration of monitoring basic data for soil and water conservation engineering measures, this study was unable to conduct a thorough investigation into the contribution rates of different types of water conservation measures that promote runoff and sediment reduction. Additionally, it did not quantitatively analyze the impact of extreme rainfall events on changes in runoff and sediment.

The Wuding River Basin lies in a transitional zone, transitioning from arid to semiarid to semihumid regions, which renders the ecological environment extremely delicate. As a result, the pressure exerted by human activities on the ecological environment is escalating. With the ongoing development and progress of the social economy within the basin, the need to strike a balance between the protection of the ecological environment and high-quality economic development has become increasingly prominent. At present, human activities that significantly influence runoff and sediment production in the basin include the construction of check dams, reservoirs, terraced fields, and the conversion of farmland to forests and grasslands. Consequently, future research should focus on investigating the effects and evolutionary trends of different types of soil and water conservation measures in the basin, unveiling profound underlying issues, and offering decision-making support to enhance the effectiveness and sustainability of various soil and water conservation measures in the Wuding River Basin.

The frequent occurrence of extreme rainfall in the Loess Plateau raises important questions that must be addressed in research on the ecological management of soil and water conservation in the Yellow River Basin. It is crucial to determine the resilience of soil and water conservation engineering in the face of extreme rainfall and the potential benefits it can provide during such events. In the middle reaches of the Yellow River, the effectiveness of soil and water conservation measures under extreme rainfall has been a topic of debate. According to some researchers [47], the sediment reduction benefits of soil and water conservation measures in most basins of the Loess Plateau exceed 80%, reaching as high as 96% in some cases, even in the presence of severe rainfall events. However, numerous other researchers argue that the benefits of watershed soil and water conservation under extreme rainfall conditions are limited and may even be negative [48]. Therefore, future studies are urgently needed to address this issue and conduct comprehensive assessments of the sediment reduction benefits of soil and water conservation measures in the Wuding River Basin during extreme rainfall conditions.

## 6. Conclusions

The Wuding River Basin, located in the middle reaches of the Yellow River, serves as the subject of investigation in this case study. The study utilizes measured data on runoff and sediment from the period spanning 1960 to 2020, aiming to analyze the characteristics of variability and investigate the underlying causes of changes in runoff and sediment. In

addition, the study employs quantitative methods to assess the contribution rates of climate change and human activities to these changes and to explore the impact of human activity factors on runoff and sediment. The main findings of this study reveal differences in the periodic characteristics of annual runoff and annual sediment transport in the Wuding River Basin. The variation period of annual sediment transport (30 years) is slightly longer than that of annual runoff (28 years). Human activities play a dominant role in reducing sediment and water in the Wuding River Basin. Grass planting and afforestation contribute significantly to the decrease in runoff, while large-scale siltation dam construction plays a major role in reducing sediment transport. The impact of climate change on sediment transport in the basin is relatively weak. NDVI is significantly negatively correlated with both runoff and sediment transport, while cultivated land area shows a positive correlation with runoff. The continuous increase in forest and grass vegetation results in more precipitation being intercepted by the vegetation canopy and the dense litter intercepting and slowing down the flow rate, with a small amount of precipitation replenishing groundwater, leading to a continuous decrease in runoff. The water storage and soil conservation effect of vegetation continues to increase, resulting in a more significant reduction in sediment. The specific conclusions are as follows:

(1) The annual precipitation in the Wuding River Basin from 1960 to 2020 exhibits a slight upward trend. Due to the implementation of large-scale soil and water conservation measures, both annual runoff and sediment transport show a significant downward trend. Notably, there are distinct change points in both annual runoff and sediment transport, occurring in 1971 and 1979, respectively.

(2) The double cumulative curves reveal a turning point in approximately 1972 and 2000. The decline in sediment transport is much more pronounced than that of runoff, and there is a noticeable decrease in sediment concentration. Comparing the reference period of 1960 to 1972 (Period I), the distribution of monthly runoff tends to be more uniform during the years 1973 to 2001 (Period II) and 2002 to 2020 (Period III), while sediment transport becomes more concentrated during the flood season. Moreover, the joint distribution of the runoff–sediment relationship also undergoes changes before and after these periods.

(3) In comparison to the reference period, climate change contributed 11.94% and −14.5% to runoff and sediment transport during Period II, respectively. Meanwhile, human activities contributed 88.06% and 114.5% to the same parameters. For Period III, climate change's contributions to runoff and sediment transport were −11.9% and −17.7% respectively, while human activities contributed 111.9% and 117.7% to the same parameters. Human activities are predominantly responsible for reducing sediment and water, with a particularly significant impact on sediment transport reduction. In contrast, the impact of climate change on sediment transport in the basin is relatively weak.

(4) The factors that exert the greatest influence on runoff are the NDVI and the area of grassland, with IncMSE values of 14.63% and 14.26%, respectively. Additionally, the area of forest contributes 6.90% and the area of cultivated land contributes 5.93%. In terms of sediment transport, the most significant human activity factors are the NDVI and forest area, with IncMSE values of 15.93% and 14.97%, respectively. Furthermore, the area of soil and water erosion control contributes 6.48% and the grassland area contributes 6.07%.

In future research, there is a need to further collect and filter natural and socio-economic indicators. Additionally, it is necessary to innovate and optimize research methods in order to explore and establish the coupling relationship between "hydro-sediment-ecological-economic". This will provide a better theoretical basis and methodological support for ecological conservation and sustainable social development in the Wuding River Basin.

**Supplementary Materials:** The following supporting information can be downloaded at: https://www.mdpi.com/article/10.3390/w16010026/s1, Figure S1: A prolonged artificial nighttime-light map of Wuding River Basin from 1995–2015; Table S1: Annual rainfall, discharge and sediment load from 1960–2020; Table S2: Water use, population and GDP of Yulin city from 1991–2015.

**Author Contributions:** Conceptualization, data curation, investigation, methodology, formal analysis, writing—original draft, writing—review and editing: J.Y.; conceptualization, methodology, supervision: Z.L.; formal analysis, validation, writing—review and editing: B.Z.; validation, project administration, funding acquisition: P.X.; validation, supervision: P.Z.; investigation, validation, writing—review and editing: J.W., W.S., S.M. and Y.Z. All authors have read and agreed to the published version of the manuscript.

**Funding:** This research was supported by the National Key Research and Development Program of China (Grant No.: 2022YFF1300805), the National Natural Science Foundation of China Major Projects (Grant No.: 42041006), the National Natural Science Foundation of China (Grant No.: U2243210), and the Science and Technology Development Foundation of Yellow River Institute of Hydraulic Research (Grant No.: HKF202312).

**Data Availability Statement:** The datasets generated during the current study are available from the corresponding author and Supplementary Materials upon reasonable request.

**Conflicts of Interest:** The authors declare no conflict of interest.

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
