# Peer review of "Transformative Trends in Runoff and Sediment Dynamics and Their Influential Drivers in the Wuding River Basin of the Yellow River: A Comprehensive Analysis from 1960 to 2020"

_water, doi:10.3390/w16010026_

Round 1

Reviewer 1 Report

Comments and Suggestions for Authors

Thank you for the presentation of your investigation. 

The article is concerned with the question of climate and human impact on the run-off from precipation in the Wudin river system, analysing the data from observations made in the field and developing a model out of that. It's an original work relevant to the field and addresses the lack of a run-off model for this are. It adds an extensive overview on different approaches in the data analysis and offers a contribution to the attribution of the different observed river behaviors. As explained in the original review I do not have any recommendations on how to improve the methodology and rather consider it complete for the purpose of the study. Of course additional investigations could be made in a follow-on study exploiting the results from the present study. The conclusions drawn in the paper appear to me to be consistent and do address the raised question on the identification of contributing factors in the river run-off.

I did not find larger issues with it. Only minor points: On page 6, Line 233, I think it should read equation 6 rather than equation 3. And in Figure 2 the printed equations are not explained and do not match anything in the text as far as I can see.

The introduction is a little bit repetitive in my opinion. But overall the presentation looks fine to me.

Author Response

Dear Reviewer 1,

We gratefully thank you for the time spent making your constructive remarks and useful suggestions, which have significantly raised the quality of the manuscript and have enabled us to improve the manuscript. Each suggested revision and comment, brought forward by the reviewer was accurately incorporated and considered. Below the comments of the reviewers are responses point by point and the revisions are indicated.

We gratefully appreciate your valuable comments. We've taken your advice completely and totally understand your concern.

  • Your comments are correct. We have corrected equation “3” to equation “6”.Please check Page 7, Line 254.
  • After revisiting our manuscript, we agree with you that removing these printed equations in Figure 2 would not have any effect on the content of our article, so we have recreated Figure 2 with these equations removed. Please check Page 11, Figure 2.
  • We have reworked the introduction section of the manuscript to make it more concise. Please check Page 2, Lines 64-69.

Reviewer 2 Report

Comments and Suggestions for Authors

This is an interesting paper that uses multiple methods of time series analysis to examine changes in annual discharge and sediment load in a major tributary of the Yellow River.

The results of the time series analysis are clearly presented. A comment on the usefulness of the various techniques could improve the paper (the change point analysis,  double mass curves, and wavelet analysis looked the most useful)

The details of the Random Forest model and the generation of the hierarchy of parameters that influenced the time series characteristics were less clearly presented and this could be improved.

There were some formatting issues (associated with tables etc.) in the version of the paper that I read.

Author Response

Dear Reviewer 2,

We gratefully thank you for the time spent making your constructive remarks and useful suggestions, which have significantly raised the quality of the manuscript and have enabled us to improve the manuscript. Each suggested revision and comment, brought forward by the reviewer was accurately incorporated and considered. Below the comments of the reviewers are responses point by point and the revisions are indicated.

We gratefully appreciate your valuable comments. We've taken your advice completely and totally understand your concern.

  • We provide a more detailed description of the details of the random forest model and the generation of the hierarchy of parameters that influenced the time series characteristics. Please check Page 10, Lines 372-405.
  • We have carefully revised all formatting issues related to tables, etc., that appear in the manuscript. Please check Page 6, Lines 208-216, Lines 232-236.

Reviewer 3 Report

Comments and Suggestions for Authors

Comments on the Quality of English Language

Author Response

Dear Reviewer 3,

We gratefully thank you for the time spent making your constructive remarks and useful suggestions, which have significantly raised the quality of the manuscript and have enabled us to improve the manuscript. Each suggested revision and comment, brought forward by the reviewer was accurately incorporated and considered. Below the comments of the reviewers are responses point by point and the revisions are indicated.

We gratefully appreciate your valuable comments. We've taken your advice completely and totally understand your concern.
